# Potential public health impacts of gonorrhea vaccination programmes under declining incidences: A modeling study

Lin Geng[1,☯], Lilith K. Whittles[2,☯], Borame L. Dickens[3], Martin T. W. Chio[4], Yihao Chen[1], Rayner Kay Jin Tan[3], Azra Ghani[2], Jue Tao Lim[1*]

**1** Lee Kong Chian School of Medicine, Nanyang Technological University, Singapore, Singapore,
**2** Department of Infectious Disease Epidemiology, Imperial College London, London, United Kingdom,
**3** Saw Swee Hock School of Public Health, National University of Singapore, Singapore, Singapore,
**4** National Skin Centre, Singapore, Singapore

☯ These authors contributed equally to this work.
* juetao.lim@ntu.edu.sg

## Abstract

### Background

Gonorrhea is the second most common sexually transmitted disease notified in Singapore in 2023. Evidence suggests that the 4CMenB vaccine designed to protect against *Neisseria meningitidis* infection may offer partial cross-protection against gonorrhea. This generated interest in using 4CMenB for the purpose of staving gonorrhea transmission. We explored the efficacy of potential gonorrhea vaccination strategies in the context of historically declining gonorrhea incidence.

### Methods and findings

We employed an integrated transmission-dynamic model, calibrated using Bayesian methods to local surveillance data to understand the potential public health impact of 4CMenB in reducing gonorrhea acquisition and transmission in men who have sex with men (MSM) in Singapore. We explored the efficacy of implementing six vaccination programmes: (**1**) offering vaccination to all male adolescents in schools (vaccination before entry [VbE]), (**2**) offering vaccination to individuals attending sexual health clinics for testing (vaccination on attendance [VoA]), (**3**) offering vaccination to individuals attending sexual health clinics and who were diagnosed with gonorrhea (vaccination on diagnosis [VoD]), or (**4**) vaccination according to risk (VaR), by offering vaccination to patients who were diagnosed with gonorrhea plus individuals who tested negative, but report having more than five sexual partners per year. We further examined how altering (**5**) VoA and (**6**) VoD strategies changed if the strategies only targeted high risk groups (VoA(H),VoD(H)). We assessed efficacy by examining vaccination impact relative to no vaccination and when behavioral parameters were held constant. We further ascertained the effects of varying vaccine uptake (10%, 33%, 100%), vaccine efficacy (22%, 31%, 47%), and duration of protection (1.5, 4, 7.5 years) on the effectiveness of each vaccination strategy.

**Data availability statement:** All code and data used for this research can be accessed at https://github.com/lineliott/gonovaxSG/.

**Funding:** This work was supported by Nanyang Technological University, Singapore—Imperial Research Collaboration Fund (INCF-2023-007 to JTL and AG). LJT is Assistant Professor at Nanyang Technological University. AG is Professor and LKW is Lecturer at Imperial College London. The funders had other role in study design, data collection and analysis, decision to publish, or preparation of the manuscript.

**Competing interests:** The authors have declared that no competing interests exist.

**Abbreviations:** CrI, credible interval; DSC, department of sexually transmitted infections control; MSM, men who have sex with men; OMVs, outer membrane vesicles; STIs, sexually transmitted infections; VaR, vaccination according to risk; VbE, vaccination before entry; VoA, vaccination on attendance; VoD, vaccination on diagnosis.

For a hypothetical 10-year vaccination programme, VbE had 14.18% of MSM gonorrhea cases averted over the time the programme was implemented. VoA had the highest protective impact on the MSM population with 40.26% averted cases (95% credible interval (CrI): 18.32%–52.57%), but required more vaccine doses than any other strategy. VoD had a smaller impact (12.04% averted cases (95% CrI: 7.12%–15.00%)), but was three times more efficient than VoA in terms of averted cases per dose. VoA(H) and VoD(H) improved the efficiency of VoA and VoD strategies by increasing averted cases per dose to 0.22 and 0.24 respectively, but conferred similar protective effects as VoA (VoA(H): 40.10% averted cases (95% CrI: 18.14%–52.55%)) and VoD (VoD(H): 12.04% averted cases (95% CrI: 7.12%–15.00%)), respectively. VaR (40.10% averted cases (95% CrI: 18.14%–52.55%)) had almost the same impact as VoA, but was more efficient by requiring administration of fewer doses than VoA, with 0.21 (95% CrI: 0.12–0.27) averted cases per dose. Sensitivity analyses indicated that VaR had the greatest public health impact with the highest number of averted cases per dose for vaccines of any efficacy or duration of protection (or both), although VoD and VoD(H) saved more vaccine resource and had the highest number averted MSM cases per dose for highly protective vaccines of long protection.

## Conclusions

Vaccination of MSM against gonorrhea, according to risk in sexual health clinics in Singapore, can be considered to reduce gonorrhea acquisition and transmission. Development of gonorrhea-specific vaccines which focuses on protective efficacy and the implementation of efficient vaccination programmes can maximize public health impact.

### Author summary

- This article explores the efficacy of potential gonorrhea vaccination strategies in the context of historically declining gonorrhea incidence, using an integrated transmission-dynamic model, calibrated using Bayesian methods to local surveillance data

- Vaccination of high risk groups was projected to have the greatest public health impact with the highest number of averted cases per dose for vaccines of any efficacy or duration of protection (or both), but vaccination after diagnoses, or vaccination after diagnoses in high-risk group individuals saved more vaccine resource and had the highest number of averted cases per dose for highly protective vaccines of long protection. Gonorrhea-specific vaccines may confer greater public health impacts by maximizing efficacy over the duration of protection under modeled vaccination programmes.

- **What do these findings mean?** Vaccination of MSM against gonorrhea according to risk in sexual health clinics can be potentially efficient and confer high public health impacts, in settings of declining gonorrhea incidence, such as Singapore.

## Introduction

*Neisseria gonorrhoeae*, a bacterial pathogen, is primarily transmitted through unprotected vaginal, anal, or oral intercourse. The infection typically affects the mucosal epithelium of male urogenital tracts, the rectum, pharynx, or conjunctiva, manifesting with inflammation, discharge, and a burning sensation during urination [1]. Among them, most male urogenital infections are symptomatic while rectal and pharyngeal infections are asymptomatic [1,2].

According to the Department of Sexually Transmitted Infections Control (DSC), Singapore, the majority of men who have sex with men (MSM) gonorrhea cases have single site infections at urethra, but up to 6% have 2 or 3 sites infections with the majority being asymptomatic (pharynx and rectal).

Globally, gonorrhea ranks among the top five most frequently reported curable sexually transmitted infections (STIs) and causes significant public health burdens [1]. While gonorrhea incidence has moderated during the COVID-19 pandemic, which may be due to reductions in sexual contacts during lockdowns/stay-at-home orders and/or reduced testing, evidence showed that gonorrhea transmission may be surpassing pre-COVID-19 levels, due to relaxation of non-pharmaceutical measures motivated by COVID-19 [3]. However, in Singapore, low but persistent gonorrhea incidence has been reported over the last few decades, with a historical decline in incidence reported in the 2000s [4]. Yet, gonorrhea remains one of the top STIs diagnosed in MSM at the DSC clinic in 2023.

The disease presents a challenge due to the ability of *N. gonorrhoeae* to evade the host's immune response, complicating pharmaceutical prevention strategies. Therefore, the primary approach to mitigating gonorrhea involves antibiotic therapy for infected individuals. However, antimicrobial resistance in *N. gonorrhoeae* has historically existed and has escalated globally [2], and the first local case of resistance was reported in 2018 [5]. Since then, the local antibiogram of gonorrhea cultures has been closely monitored for decreased susceptibility and resistance to the recommended antibiotics, but to date, few resistant strains have been diagnosed in Singapore. However, as there are limited antibiotics to target gonorrhea, the exploration of more sustainable interventions such as vaccination for public health preparedness remains crucial [6].

The interest in vaccine prevention against *N. gonorrhoeae* arose from observations of declining gonorrhea incidence with the use of the MenB (one type of meningococcal serogroups) outer membrane vesicles (OMVs) vaccine in Norway [7] and Cuba [8,9], which were used to protect against meningococcal disease caused by meningococcal group B bacteria. These observations suggested that meningitis prevention might also protect against gonorrhea. A retrospective observational case-control study confirmed that the *Neisseria meningitidis* serogroup B OMV vaccine (MeNZB [GSK]) provided a 31% (95% credible interval (CrI) 21–39) efficacy against gonorrhea with a 3-dose program [10]. Subsequent studies provided further genetic evidence that the anti-gonococcal antibodies induced by MeNZB can offer additional cross-protection against gonorrhea [11]. Furthermore, studies from the United States, Canada, and Australia have reported that 4CMenB, an alternative vaccine option for gonorrhea, can be more effective than MeNZB, with efficacies ranging between 33% and 46% [12]. However, a recent study suggested a more conservative, statistically insignificant 22% efficacy of 4CMenB vaccine in gonorrhea prevention, which may be due to an underpowered sample size [13].

Prior modeling work has also highlighted the potential cost-effectiveness of introducing the 4CMenB vaccine in the United Kingdom through various uptake strategies, advocating for vaccination according to risk (VaR) among the MSM population. This approach is projected to avert an average of 110,200 cases (95% CrI 36,500–223,600) over 10 years in the study setting [14]. However, further analysis suggested that pessimistic vaccine sentiments may attenuate the benefits of these vaccination programmes [15].

Nevertheless, current work has focused on high and/or increasing transmission settings for gonorrhea. In contrast, Singapore's epidemiology presents a unique opportunity to assess the population health impacts of gonorrhea vaccination programmes in the context of historically declining incidence over the past 2 decades. While the previous climate of criminalized same-sex relationships has hindered the gathering of statistics in MSM populations,

decriminalization also provided a rare opportunity to study and prevent STIs in a group that has been disproportionately affected globally. In this study, we aimed to guide potential public health decisions regarding the implementation of different gonorrhea vaccination programmes. We sought to devise and compare alternative, pragmatic approaches to implementing vaccination within sexual health clinics or educational institutions, in terms of their public health impact and efficiency. To do so, we used a high-resolution compartmental model of gonorrhea and utilized a Bayesian evidence synthesis framework to characterize the transmission dynamics of gonorrhea. This was calibrated using three different case-time series of gonorrhea incidence among MSM from 2004 to 2018. We examined how the impact of vaccination programs over 2024–2034 depends on vaccine-specific factors, such as uptake, efficacy, and duration of protection, as well as different epidemic trajectories.

## Methods

### Model structure

We adapted a previously published compartmental model to characterize gonorrhea transmission among the MSM population in Singapore [14]. The model captured how individuals move between uninfected, incubating, asymptomatic/symptomatic and treatment stages. Vaccination status was incorporated in terms of unvaccinated, vaccine-protected and those with waned vaccine protection, as well as low- and high-risk sexual activity groups (Fig 1).

### Data

The model was calibrated using a Bayesian inference framework using proxies of annual gonorrhea incidence in the MSM population from 2004 to 2018 in Singapore. MSMs were defined to be men who report ever having sex with a man. As actual breakdowns of annual gonorrhea MSM incidence were not available in the study setting, we assumed that this was proxied by (**1**) male incidence minus the female incidence of gonorrhea, (**2**) the male incidence of gonorrhea, i.e., the upper bound to the annual gonorrhea MSM incidence, and (**3**) the male incidence multiplied by the proportion of reported gonorrhea cases in the MSM population in other study settings (See Table A1 in S1 Text). Raw annual gonorrhea incidence was obtained from the communicable disease surveillance system of Singapore from 2004 to 2020 [4] and excluded observations in 2019, 2020 due to COVID-19. Publicly available estimates for gonorrhea incidence are not available beyond 2020. We calibrated our model under each scenario and reported model outputs based on the first scenario as our main results.

### Model calibration

Markov chain Monte Carlo was used to infer parameters of the model, which included parameters characterizing gonorrhea prevalence, screening behavior, and transmission parameters. We used 500,000 iterations and 8 chains in parallel to sample the posterior distribution of parameters. The full specification of the model is detailed in the S1 Text (Section 1). Convergence was assessed using visual inspection of trace plots, the Gelman-Rubin statistic, and the effective sample size. One thousand random parameter sets were sampled from the overall sampling distribution to characterize inherent parameter uncertainty for use of simulating vaccination programmes.

### Vaccination scenarios

Post-model calibration, we randomly sampled 1,000 sets of parameters from the estimated posterior distributions. This was done to simulate the baseline case of having no vaccination

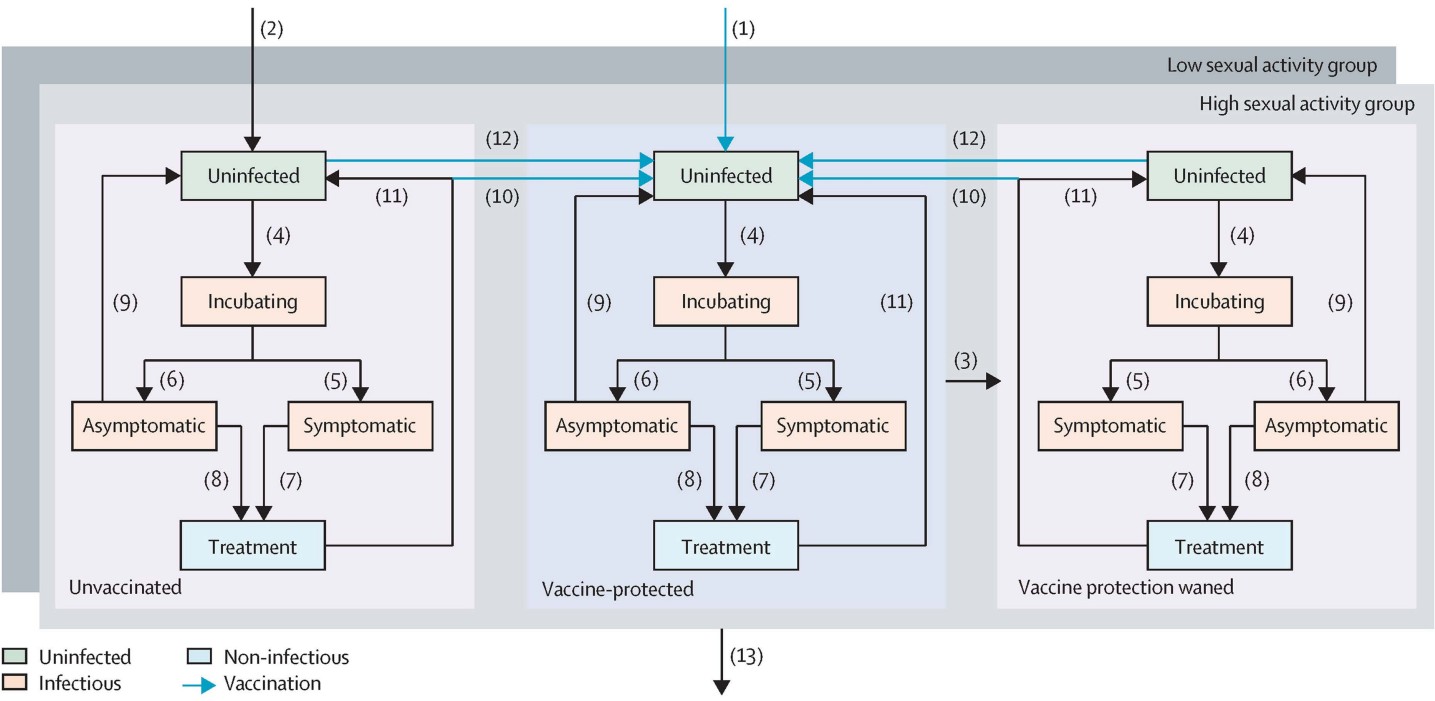

**Fig 1. Model structure diagram of gonorrhea transmission in different sexual activity groups.** The model categorizes individuals into uninfected, incubating, infected (asymptomatic or symptomatic), and treated compartments. The model characterizes a constant population over time by incorporating individuals who enter (1, 2) and leave (13) sexually active age. A certain proportion of the population is initially set to be infected, while the remainder were deemed uninfected. Uninfected individuals (green compartments) who have had sexual contact with infected individuals may be exposed to and get infected with gonorrhea (4). Those in the incubating stage will develop either symptomatic (5) or asymptomatic (6) infection, and they can spread gonorrhea to the uninfected population through sexual contact (orange compartments). Individuals with either symptomatic or asymptomatic infection are assumed to seek treatment or regular sexually transmitted infection (STI) screening from sexual health clinics (7, 8) with some probability. However, individuals with asymptomatic infections can recover naturally (9). Both asymptomatic and symptomatic infections are diagnosed through laboratory tests, and can be cured from treatment and remain non-infected shortly (blue compartments) before returning to the susceptible compartment (11). Vaccination is offered in school-based programme aimed at male teenagers entering sexually active age (1), or upon diagnosis (11) and/or screening (12) in sexual health clinics, based on different vaccine administering programme (blue arrows). Identical arrangements of gonorrhea transmission-vaccination processes are specified for both high and low sexual activity groups (light and dark gray panel), while accounting for heterogeneity in epidemiological parameters in both groups.

programmes implemented. Here, we assumed either that (**1**) gonorrhea transmissibility remained stable over time or (**2**) gonorrhea transmissibility had a linearly decreasing trend.

We considered vaccination of 4CMenB by offering vaccinations to all male adolescents as a nationwide school vaccination programme before they reach sexually active age (15 years old, vaccination before entry [VbE]) and offering of vaccination to MSM when they attend sexual health clinics for screening (vaccination on attendance [VoA]) or when they are diagnosed with gonorrhea at a clinic (vaccination on diagnosis [VoD]). We also explored offering vaccination to high-risk individuals (VaR), taking five partnerships per year as the threshold to categorize individuals into the high-risk sexual activity group, following prior work [14]. We note that the VaR strategy precluded offering vaccination to patients who are both at low risk and/or not diagnosed upon their attendance to the health clinic. We also modified VoA and VoD strategies by offering vaccinations in these groups to only high-risk individuals (VoA[H]), VoD[H]). Table 1 provides a summary of each simulated vaccination program.

To quantify the raw public health impact of each vaccination strategy, we reported the

(1)  Averted number of cases: computed as the difference in the number of treated gonorrhea patients between the baseline case and the respective vaccination strategy.

**Table 1. Target populations for different vaccination strategies.**

|  | Schools | Attending sexual clinics | | | |
|  | All adolescents (15-year-olds) | On screening | | On diagnosis | |
|  |  | At low risk | At high risk | At low risk | At high risk |
|---|---|---|---|---|---|
| VbE | √ |  |  |  |  |
| VoD |  |  |  | √ | √ |
| VoD(H) |  |  |  |  | √ |
| VoA |  | √ | √ | √ | √ |
| VoA(H) |  |  | √ |  | √ |
| VaR |  | √ | √ | √ |

We reported programme efficiency by computing the

(2) Percentage of averted cases: computed as the percentage difference in the number of treated gonorrhea patients between the baseline case and the respective vaccination strategy.

(3) Averted cases per dose: computed as the difference in the number of treated gonorrhea patients between the baseline case and the respective vaccination strategy, divided by number of vaccination doses given.

We also computed vaccination strategy-specific metrics to understand the resource needs and uptake profile for each strategy:

(4) Number of vaccinations administered: measured by the number of times vaccination was provided. We assumed that all individuals who accepted vaccination took two doses for primary protection and one booster dose for revaccination.

(5) Primary uptake ratio: which tracked the proportion of individuals who received vaccine protection for the first time given the number of vaccinations.

## Vaccine profile and uptake

We used the efficacy of 4CMenB at 31% (95% CrI: 21–39) as the primary vaccine profile to run simulations of each vaccination programme [10]. We assumed that two doses are required to confer the full protective efficacy, and assumed that individuals who take the first dose will take the second dose, to reach the overall uptake rate of 33% in sexual health clinics for both primary protection (two doses) or revaccination (one dose) in MSM population [16] and 87.3% in schools [17]. Uptake rates were compiled respectively based on prior information on uptake rates of vaccination in schools/sexual clinics for other similar vaccination programmes (Table 2).

While the duration of protection of 4CMenB against gonorrhea is still unclear, observations reported high human complement serum bactericidal antibody assay tilters against fHbp, NadA, and PorA on adolescents for 18–36 months [18], young adults for 4–7.5 years [19] after two dose of primary 4CMenB uptake, suggesting the vaccination with 4CMenb can confer humoral immunity against gonorrhea for around 4 years, which we used as our central estimate. Table 2 summarizes the vaccine profiles used for simulating each vaccination programme.

## Sensitivity analyses

Sensitivity analyses was also conducted to examine the potential impact of varying vaccine profiles and vaccination programme-related parameters on each programme's population

Table 2. Vaccination related parameters used for simulating vaccination programme.

| | Definition | Main | Alternative scenarios |
|---|---|---|---|
| $e$ | Vaccine efficacy | 31% [10] | 22% [13], 31% [10], 47% [20] |
| $D_v$ | Protection of duration | 4 years [19] | 1.5 [18], 4 [19], 7.5 [19] years |
| $p^{VbE}$ | Uptake rate of adolescent vaccination | 87.3% [17] | 10%, 33% [16], 100% |
| $p_j^{VoD}$ | Uptake of vaccination on diagnosis by infected in group $j$ | 33% [16] | 10%, 33% [16], 100% |
| $p_j^{VoS}$ | Uptake of vaccination on screening by uninfected in group $j$ | 33% [16] | 10%, 33% [16], 100% |

health impact and efficiency. We varied the protective efficacy of the vaccine from 22%, 31%, and 47% to reflect mild, normal, and strong vaccine efficacies, respectively, as well as setting the duration of protection to 1.5 years, 4 years, and 7.5 years to reflect short, normal, and long duration scenarios, respectively. We further varied vaccination uptake rates to 10%, 33%, and 100% to reflect low, central, and high uptake scenarios.

The public health impact of each vaccination program was assessed relative to the baseline case of no vaccination. In our analyses, we kept vaccine-related parameters stable across the entire period of implementing each vaccination program.

We separately calibrated the transmission models based on the different proxies of MSM gonorrhea incidence (see Data section). We projected the future incidences based on constant or linearly trending transmission rates over 2024–2034 and assessed the influence of these separate models on vaccination programmes' public health impacts over the projected time horizon. Furthermore, in order to ascertain how the uncertainty in different parameters would contribute to the overall uncertainty of the results, we iteratively set each parameter by its median values and all other parameters to vary according to the sampled distribution. Thereafter, each of the vaccination programmes' impacts over a 10–30-year timeframe was projected, under the three separate case series.

We further assessed the predictive ability of our models through an expanding cross-validation scheme, where the data was split into initial testing and training sets (70%, 30%), and the model was calibrated to the training set and annual gonnorhoea incidence rates projected forward in time to assess concordance with the test-set observations. In the subsequent cross-validation steps, an additional year of data was added and the procedure is repeated until the testing set is depleted. Results were inspected visually to determine whether the model could capture the general trends in annual gonorrhea incidence rates in the test set, and quantified through computation of mean absolute percentage errors.

Lastly, we examined the sensitivity of behavioral and epidemiological parameters on our model's outputs of the prospective vaccination strategies efficiency and raw public health impacts. This was done by varying the parameters (i.e., screening rates, recovery rates, asymptomatic rates, proportion of individuals in high/low risk groups) by ±25,50% while keeping all other parameters fixed where possible, and reprojecting each vaccination strategy through 2024–2034.

All code and data used for this research can be accessed at https://github.com/lineliott/gonovaxSG/.

## Results

In the baseline case of no intervention, from 2024 to 2034, there will be an estimated 71,669 (95% CrI: 36,926–100,062) gonorrhea cases among the MSM population in Singapore, with

an estimated 16.53% (95% CrI: 12.08%−20.89%), receiving treatment at sexual health clinics. 16.85% (95% CrI: 4.75%−27.76%) of treated infected persons are estimated to be asymptomatic, and 83.15% (95% CrI: 68.34%−92.50%) are symptomatic. An estimated 145,449 (95% CrI: 32,325−248,501) will be screened where the high-risk group comprised 24.60% (95% CrI: 14.67%−40.21%) of all screened individuals (Table 3).

Vaccination strategies (Table 1) were simulated forward in time and compared to the baseline case of no interventions. In our primary results, we presented the conservative scenario where gonorrhea transmission stabilized at 2018 levels. Whereas we described the case where gonorrhea transmission in 2024−2034 followed the historically declining trends from 2004 to 2018 as an alternative baseline in the S1 Text (Section 5). We used the 4CMenB vaccination profile (Table 2) to simulate each vaccination strategy (Fig 2).

For the vaccination on entry (VbE) strategy, adolescent males received vaccination at an uptake rate of 87.3%, resulting in a maximum primary uptake ratio of 100% (Table 4). However, this strategy had a low impact by averting only 14.18% (95% CrI: 9.11%−17.08%) of cases, but required administration of 172,701 vaccinations. This strategy was also not efficient and only yielded 0.005 (95% CrI: 0.003−0.006) averted cases per dose.

We also simulated vaccinating individuals on diagnosis (VoD) or on attendance (VoA). VoA had a comparatively higher number of vaccinations administered at 45,699 (95% CrI: 13,004−73,327), and a primary uptake ratio of 91.62% (95% CrI: 85.35−95.57), compared to VoD, at 3,317 (95% CrI: 1853−4,252) vaccinations administered and a primary uptake ratio of 97.13% (95% CrI: 95.63−97.96). This consequentially led to VoA and VoD achieving 40.26% (95% CrI: 18.32%−52.57%) and 12.04% (95% CrI: 7.12%−15.00%) averted cases, respectively. This yielded 0.06 (95% CrI: 0.02−0.10) and 0.22 (95% CrI: 0.12−0.28) averted cases per dose for VoA and VoD strategies, respectively. As the majority of attendances involved the low-risk group, while most diagnoses came from the high-risk group (Table 3), VoA had a broader impact by covering both low- and high-risk individuals through screening, whereas VoD primarily targeted high-risk individuals upon diagnosis. Therefore, VoA had a greater raw population health impact, in terms of higher percentage of averted cases, but VoD was a more efficient strategy with a larger number of averted cases per vaccination dose.

**Table 3. Summary of number of individuals transitioning to each compartment in the transmission model by risk group from 2024 to 2034. Results presented are relative to the main baseline scenario, where no vaccination programmes were implemented. Transmission probabilities of 2024–2034 were kept constant at 2018 levels.**

| Category | High risk | Low risk | Total |
|---|---|---|---|
| Screened | 33,900 (9,070–51,784) | 111,549 (20,958–202,986) | 145,449 (32,325–248,501) |
| Screened (%) | 24.60 (14.67–40.21) | 75.40 (43.56–84.11) | |
| Incidence | 63,399 (32,708–87,539) | 8,270 (46–16,115) | 71,669 (36,926–100,062) |
| Incidence (%) | 88.62 (81.32–99.30) | 11.38 (0.07–18.32) | |
| Treated | 10,460 (5,660–14,155) | 1,250 (8–2,346) | 11,709 (6,327–15,333) |
| Treated (%) | 89.37 (82.22–99.32) | 10.63 (0.07–17.42) | |
| Asymptomatic and diagnosed | 1,753 (429–3,058) | 124 (0–307) | 1878 (470–3,199) |
| Asymptomatic diagnosed (%) | 93.35 (84.06–99.56) | 6.65 (0.04–13.45) | |
| Symptomatic and diagnosed | 8,706 (4,600–11,839) | 1,125 (7–2,148) | 9,832 (5,137–13,140) |
| Symptomatic and diagnosed (%) | 88.62 (81.32–99.30) | 11.38 (0.07–18.32) | |
| Treated/Incidence (%) | 16.68 (12.25–21.06) | 15.45 (11.08–19.88) | 16.53 (12.08–20.89) |
| Asymptomatic and treated (%) | 16.85 (4.75–27.76) | 10.18 (1.69–19.76) | 16.14 (4.41–26.73) |
| Symptomatic and treated (%) | 83.15 (68.34–92.50) | 89.82 (76.87–96.73) | 83.86 (69.48–93.06) |

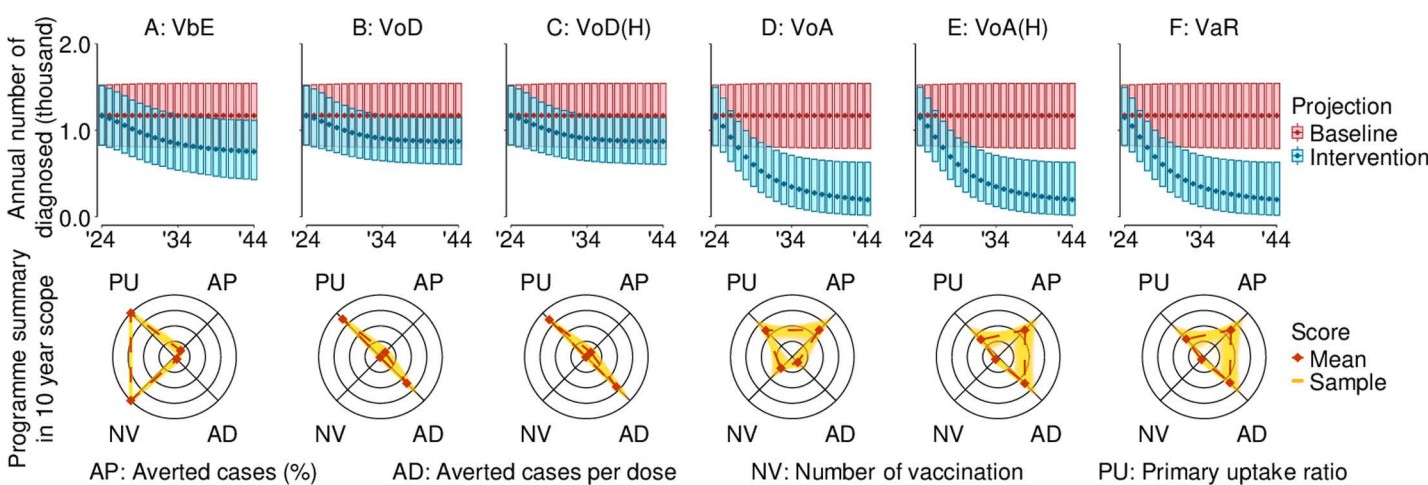

**Fig 2. Projected number of annual diagnosed cases by vaccination programme over 2024–2034.** The posterior mean is marked with a square, and the 95% credible interval is indicated by the boundaries of the box. Projections assume that transmission probabilities from 2024 to 2034 were kept constant at 2018 levels. One thousand posterior draws were used to generate future projections and calculate programme-specific metrics. The series of boxplots illustrate the annual future number of diagnosed cases under different vaccination strategies compared to a no-vaccination programme baseline. Each column represents a different vaccination programme. Respectively, these are denoted **(A)** vaccination before entry (VbE), **(B)** vaccination on diagnoses with gonorrhea (VoD), **(C)** VoD strategies by offering vaccinations in these groups to only high sexual activity individuals (VoD[H]), **(D)** vaccination on attendance (VoA), **(E)** VoA strategies by offering vaccinations in these groups to only high sexual activity individuals (VoA[H]), and **(F)** vaccination according to risk (VaR). The radar graph visualizes the expected impact, efficiency, and uptake of each vaccination programme from 2024 to 2034. Metrics are averted cases by percentage (AP: computed as the percentage difference between number of treated gonorrhea patients between the baseline case and the respective vaccination strategy), averted cases per dose (AD: computed as the difference between number of treated gonorrhea patients between the baseline case and the respective vaccination strategy, divided by number of vaccination doses given), number of vaccinations administered (NV: measured by the number of times vaccination was provided. We assumed that all individuals who accepted vaccination took two doses for primary protection and one booster dose for revaccination), and primary uptake ratio (PU: tracked the proportion of individuals who received vaccine protection for the first time given the number of vaccination). Each sample's metric value is normalized by the maximum and minimum across strategies for visual clarity, points closer to the center are closer to the overall strategy minimum metric score, and the radar border represents the overall maximum score that each strategy can achieve.

**Table 4. Summary of population health impact for various vaccination programmes from 2024 to 2034.** We report posterior means and accompanying 95% credible intervals. Estimates presented are relative to the baseline scenario of no intervention, with transmission probabilities from 2024 to 2044 kept stable at 2018 levels. * Assumes that individuals received two doses for primary uptake, and one dose for revaccination.

| Strategy | Averted cases (total) | Averted cases (%) | Averted cases per dose | Number of vaccinations administered* | Primary uptake ratio |
|---|---|---|---|---|---|
| VbE | 1,648 (942–2,120) | 14.18 (9.11–17.08) | 0.005 (0.003–0.006) | 172,701 (172,701–172,701) | 100.00 (100.00–100.00) |
| VoD(H) | 1,427 (551–2,176) | 12.04 (7.12–15.00) | 0.24 (0.14–0.31) | 2,954 (1,674–3,912) | 96.79 (95.47–97.69) |
| VoD | 1,427 (551–2,176) | 12.04 (7.12–15.00) | 0.22 (0.12–0.28) | 3,317 (1853–4,252) | 97.13 (95.63–97.96) |
| VoA | 4,704 (1948–6,947) | 40.26 (18.32–52.57) | 0.06 (0.02–0.10) | 45,699 (13,004–73,327) | 91.62 (85.35–95.57) |
| VoA(H) | 4,685 (1938–6,922) | 40.10 (18.14–52.55) | 0.22 (0.12–0.28) | 11,622 (5,502–15,669) | 87.13 (79.54–92.24) |
| VaR | 4,685 (1939–6,922) | 40.10 (18.14–52.55) | 0.21 (0.12–0.27) | 11,874 (5,906–15,851) | 87.40 (79.73–92.61) |

We further simulated vaccination strategies which focused on administering vaccines based on the frequency of individuals' sexual activity. VoA(H) vaccinates every individual attending a sexual health clinic, regardless of infection status, but only if they reported high sexual activity, whereas VoD(H) vaccinates every diagnosed individual only if they reported high sexual activity. VoA(H) and VoD(H) yielded 40.10% (95% CrI: 18.14%−52.55%) and 12.04% (95% CrI: 7.12%−15.00%) averted cases, respectively, similar to VoA and VoD. However, the number of averted cases per dose for VoA(H) tripled (0.22 95% CrI: 0.12−0.28) compared to VoA. However, for VoA(H), number of individuals vaccinated was far lower in comparison to VoA, with only 11,622 (95% CrI: 5,502−15,669) number of vaccinations administered, with

a primary uptake ratio at 87.13% (95% CrI: 79.54%−92.24%). By offering vaccination only to individuals of high risk, vaccine resources could potentially be reduced and targeted towards high-risk individuals who are more likely to contribute to gonorrhea transmission.

We further simulated risk-based vaccination strategies by VaR—where vaccination was offered to all diagnosed individuals and high sexual activity attendees reporting to sexual health clinics. Here, VaR required 11,874 (95% CrI: 5,906−15,851) vaccinations, with a primary uptake ratio at 87.40% (95% CrI: 79.73%−92.61%) and yielded over 40.10% (95% CrI: 18.14%−52.55%) averted cases and with 0.21 (95% CrI: 0.12−0.27) averted cases per dose. These are almost identical to VoA(H) because the only difference in individuals offered vaccinations comprised diagnosed patients with low sexual activity whom contributing minimally to gonorrhea transmission.

## Sensitivity analysis

We presented results for sensitivity analysis on vaccination profiles and their impact on the population health impact of vaccination programmes (Fig 3). In general, better protective efficacy and duration of protection allowed vaccination strategies to confer greater population health impact. Better vaccines allowed impactful strategies (VoA, VoA(H), and VaR) to achieve superior population health impact and efficiency versus the best strategies (VoA, VoD(H)) under the standard vaccine profile. This observation supported that targeting individuals with high sexual activity when they attend for screening with a vaccine of improved profile can achieve better public health outcomes in an efficient manner versus the current standard vaccine profile (Fig 3).

Specifically, VbE, VoD, VoD(H) did not attain high population health impact, in terms of number of cases averted, and high efficiency, in terms of number of averted cases per dose,

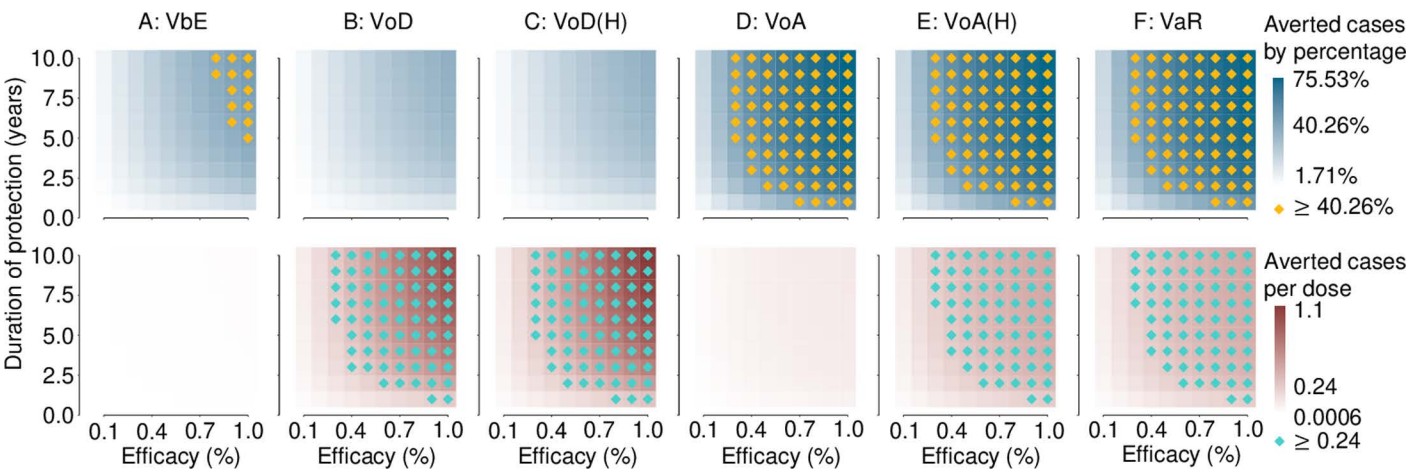

**Fig 3. Vaccine impact and efficiency under varying vaccine profiles—Sensitivity analysis from 2024 to 2034.** Results presented were relative to the baseline case of no vaccination programmes being implemented. Projections assumed that transmission probabilities of 2024–2034 were kept constant at 2018 levels. The heatmaps visualizes the efficiency (averted cases by percentage and averted cases per dose) of varying vaccine durations of protection and efficacy under different strategies. The evaluation was based on the percentage of averted cases and averted cases per dose. Each unit is color-coded from light to dark, based on the magnitude of the point estimates, under selected vaccine efficacy and duration of protection. Vaccination programmes were denoted by column and were namely: **(A)** vaccination before entry (VbE), **(B)** vaccination on diagnoses with gonorrhea (VoD), **(C)** VoD strategies by offering vaccinations in these groups to only high sexual activity individuals (VoD[H]), **(D)** vaccination on attendance (VoA), **(E)** VoA strategies by offering vaccinations in these groups to only high sexual activity individuals (VoA[H]), and **(F)** vaccination according to risk (VaR). The highlighted square represents instances where vaccination programmes yielded more than 40.26% averted cases compared to the baseline scenario or when the vaccination programme yielded more than 0.24 averted cases per dose. The threshold was drawn by referring to Table 4, which summarized the best performance for each vaccination programme for each metric, respectively, i.e., impact by VoA and efficiency by VoD(H), under the standard vaccine profile of efficacy = 31%, duration = 4 years.

even with an optimal vaccine profile (efficacy = 100% and duration = 10 years). Whereas for VoA, VoA(H), and VaR, improvements in vaccine programme impact and efficiency relied more on the vaccine's efficacy, rather than duration of protection. In terms of vaccine efficiency, we observed that VaR and VoA(H) can surpass the threshold of 0.24 averted cases per dose achieved by VoD(H) if vaccine profiles are improved. This may be due to these strategies being more efficient in targeting high sexual activity populations and consequentially being able to compensate for the loss in vaccine efficiency (Fig 3).

Simulating a more optimistic baseline case where gonorrhea transmission probabilities decreased in 2024–2034 following the trend from 2004 to 2014, and/or calibration of the transmission model to alternative time series of gonorrhea incidence in MSM, did not significantly affect our results on vaccination programme efficiency and effectiveness (See S1 Text, Section 5).

Increasing vaccine uptake also conferred greater public health impact for all strategies in terms of the percentage of cases averted (Fig 4), but at very high uptakes, the efficiency of all strategies in terms of number of averted cases per dose was reduced. Specifically, for higher uptake rates, the impact of the vaccine measured by the percentage of averted cases, always increased, but efficiency consistently decreased. However, VoD and VoD[H] showed minimal declines in efficiency even as uptake rates increased, which may be due to administration of vaccines in individuals who are at high risk even at high uptake rates. In contrast, VoA, VoA[H], and VaR demonstrated that higher uptake rates significantly reduced the efficiency of the programmes, in terms of averted cases per dose, especially with better vaccines.

Across uptake rates, higher durations of vaccine protection increased averted cases by percentage, but kept averted cases per dose similar, whereas higher vaccine efficacy

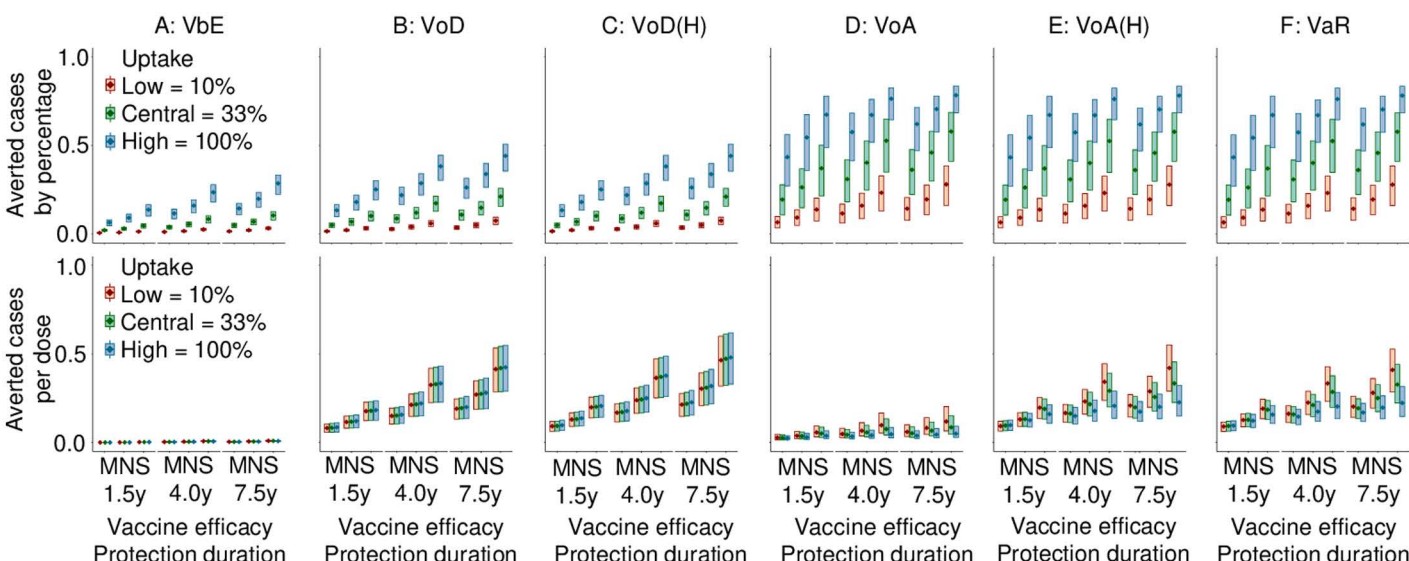

**Fig 4. Vaccine programme public health impact and efficiency under different vaccine profiles and uptake ratios.** Results presented were relative to the baseline case of no vaccination programmes being implemented. Projections assume that transmission probabilities of 2024 to 2034 were kept constant at 2018 levels. The series of boxplots illustrate the differences in vaccination strategy impact by varying vaccine uptake rates, duration of protection, and vaccine efficacy. The uptake rates were set at 10%, 33%, and 100% in both school-based and sexual health clinic vaccination programme strategies. Respectively, these are denoted **(A)** vaccination before entry (VbE), **(B)** vaccination on diagnoses with gonorrhea (VoD), **(C)** VoD strategies by offering vaccinations in these groups to only high sexual activity individuals (VoD[H]), **(D)** vaccination on attendance (VoA), **(E)** VoA strategies by offering vaccinations in these groups to only high sexual activity individuals (VoA[H]), and **(F)** vaccination according to risk (VaR). Vaccine quality was set based on clinical observations with vaccine efficacy categorized as mild (M) at 22%, normal (N) at 31%, and strong (S) at 47%, with protection durations of 1.5 years, 4 years, and 7.5 years, respectively.

increased both averted cases by percentage and averted cases per dose. For jointly higher durations of protection and higher vaccine efficacies, only averted cases per dose increased moderately (Fig 4).

By decomposing the uncertainty in our projections, we found that epidemiological and behavioral parameters, such as initial transmission rates, treatment-seeking behavior, and screening rates had a large influence on the uncertainty of the estimated impacts for the vaccination programmes, whereas parameters such as the rate of leaving the incubation period and the probability of symptomatic infection did not contribute to large variations in projected results. Simulating our results over a 30-year instead of 20-year timeframe also indicated that the hypothetical reductions in incidence can be maintained for VoA, VoA(H), and VaR around the 20-year mark and remain at a suppressed level, assuming a constant rate of transmission at 2018 levels.

## Discussion

We calibrated a transmission model of gonorrhea to explore the population health impacts of potential vaccination strategies over a 10-year period. Over 2024–2034, we found that strategies which targeted MSM on diagnoses (VoD, VoD(H)) were highly efficient in terms of number of averted cases per dose, but had less population health impact, compared to vaccination of MSM on attendance (VoA), in terms of the total number of averted cases. While these results highlighted certain trade-offs between population health impact and efficiency among vaccination strategies, we found that vaccination based on both attendance and only offering vaccines to high-risk groups or vaccination by targeting those at risk (VoA[H], VaR) can confer both the benefits of having a large number of averted cases and preserve acceptable efficiency with an estimated 0.22 (95% CrI: 0.09–0.29) averted cases per dose. This suggested that impact and efficiency gains were mainly conferred from targeting MSM with a high number of sexual partners. From an implementation standpoint, VaR may be the most practical strategy as high sexual activity MSM diagnosed with gonorrhea may not disclose partnership numbers, and may be classed as MSM at low risk and not be vaccinated under VoA(H). Conversely, these same individuals would be eligible under VaR.

Increasing vaccine uptake also conferred greater public health impact for all strategies in terms of the percentage of cases averted (Fig 4), but higher vaccine uptake was estimated to reduce the efficiency of all strategies in terms of number of averted cases per dose. Specifically, as uptake rates increased, sensitivity analyses revealed that there was always a corresponding increase in the percentage of averted cases, but was accompanied by a corresponding decline in the efficiency of the programme (in terms of averted cases per dose). However, VoD and VoD[H] showed minimal declines in efficiency even as uptake rates increased, which may be due to administration of vaccines in MSM who are at high risk even at high uptake rates. In contrast, VoA, VoA[H], and VaR demonstrated that higher uptake rates significantly reduced the efficiency of the programmes, in terms of averted cases per dose.

Our work corroborates with literature on the potential public health benefits of vaccination programmes of STIs. Gonorrhea vaccination in settings of increasing incidence were projected to be cost-effective when administered according to risk in sexual health clinics [14]. Pre-emptive vaccination was expected to reduce the likelihood of mpox outbreaks in the same study setting, especially in targeting of dense and well-connected sexual contact networks [21]. Whereas vaccination for human papillomavirus could reduce the risk of cervical cancer with increasing improvement in equality in cervical cancer incidence in the future [22].

A key strength of our analysis was that we calibrated our transmission-dynamic model to three separate case series and used Bayesian methods to account for uncertainty in epidemiological parameters. Our simulation framework directly captured the uncertainty in

these parameters, including those that characterize heterogeneities and changes in sexual behavior and testing, by using a large number of parameters drawn from the posterior distribution of data to simulate future changes in gonorrhea transmission in different vaccination programme scenarios. As the future transmission trajectory of gonorrhea is unknown, we considered alternative trajectories, such as assuming constant rates of transmission, or declining rates of transmission for 2024–2034 (See S1 Text, Section 5). We reported conservative estimates given constant rates of transmission and presented the alternative scenario in the S1 Text. We incorporated various hypothetical vaccine profiles as the vaccine efficacy and protection of duration of potential gonorrhea vaccines is currently uncertain. Our simulations also comprised vaccination programmes that can be implemented in a real-world setting, such as through sexual health clinics or schools, which further highlighted the applicability of our simulations.

There are certain limitations to this research study. In our exploration of the public health impacts of these vaccination programmes, we did not study the cost-effectiveness of these strategies, which departed from prior modeling work done in settings of historical increasing incidence [14], due to the lack of publicly available cost data on gonorrhea treatment in the local context. We did not calibrate our model to actual gonorrhea incidence in MSM populations, as these were not available. Instead, we calibrated our model to three alternative epidemiological time series which could realistically proxy epidemiological patterns in MSM populations across the years—across these different scenarios, we found that our results on vaccination programme impact and efficiency did not vary significantly. Certain parameters employed for model calibration also relied on information from other study settings, such as the natural recovery rate, and the proportion of asymptomatic infections, although there should be no major location-specific differences in these parameters. We also characterized the change in transmission probability over the observation period using a linear function, or assumed that they were constant in the future. Our exploration included an alternative exponentially decaying functional form to characterize changes in transmission probabilities, but this did not fit to observed historical trends appropriately. Additionally, we excluded data from 2019 and 2020 from model calibration, as non-pharmaceutical measures motivated by COVID-19 may have exogenous impacts on gonorrhea transmission, which cannot be captured by our model [14]. Future work should consider explicitly accounting for these factors and incorporate more recent, local epidemiological information on gonorrhea transmission. There was also a lack of community sexual behavior data, such as information on sexual networks, to characterize behaviors and proportions of key at-risk populations in the study setting. Hence, it was difficult to parameterize changes in sexual behaviors after vaccination over study period. We relied on survey data from a UK MSM demographic survey to parameterize screening rates in high/low sexual activity groups in Singapore, and explored the impact of different screening and sexual behavior parameters in sensitivity analysis. However, due to recent legislative changes in decriminalizing sex between consenting adult males in the study setting, the practicality of collecting such data in the study setting is increased, and future work should seek to ascertain potential vaccination programme effectiveness using more accurate local data, as well as understand the impact of vaccination on sexual behaviors. We hypothesize that vaccination may be accompanied by higher risk activities due to an assumption of efficacious vaccine-mediated protection, and thereby modulate the public health impact of vaccination programmes. Our analysis may also require constant updates in conjunction with rapidly developing literature surrounding the protective efficacy of 4CMenB against gonorrhea infection. The most recent systematic review reported a pooled efficacy of 32.4% [23], a finding that supported the parameters used in our primary analysis. However, a more recent retrospective cohort study reported a 23% protective effectiveness against

gonorrhea [24], highlighting heterogeneities in published estimates. We also presented a simplified analysis of vaccination programmes, where protective efficacies and duration were identical for both the primary/secondary dose and the booster dose. However, this biased our estimates of public health impact conservatively as protective efficacies should be higher when protection is conferred under the booster dose. Our model only incorporated high/low risk sexual activity groups, but assumed homogeneity in behaviors within each group. Proximal factors, such as past sexual history, unprotected sexual behavior, or substance use were also not accounted for due to lack of available granular data. Therefore, estimation of the force of infection were coarsened to either of high/low risk population subgroups. Potentially important population subgroups, such as superspreaders, were not accounted for. Higher efficiency vaccination strategies, such as targeting those at very high risk, were therefore not explored, as parameters for these population subgroups were not available for model calibration and scenario analysis. The main study results were also based on a constant population size over time, but shifts in age distribution and migration may influence our results. A decreasing population size over time would lead to a gradual depletion of susceptibles and lead to gradually lower raw public health impacts for the vaccination programmes, and vice versa, for an increasing population size. However, data on the future population age structure and numbers over the next 10–20 years is currently not available and these scenarios were precluded from our modeling study. As our output was primarily focused on projecting the potential future impacts of the vaccination programmes, it was difficult to forecast and incorporate changes in public health campaigns, changes in healthcare access, or risk perception, which may influence screening rates over the future 10–20-year time horizon. We therefore relied on a constant, risk-group-specific screening rate for these projections. Hypothetically, increases in screening rates due to intensified public health campaigns may increase the public health impacts of strategies such as VoA and VoD, by provision of vaccination to more individuals attending sexual health clinics. Combining modeled vaccination programmes with alternative public health strategies may also amplify the public health impact of the vaccination programmes. These scenarios can be explored in future studies. Utilization of rapid test kits and intensified outreach can enhance the uptake rate of vaccination programmes in at-risk populations. Mitigation of potential stigma towards gonorrhea through intensified outreach, and increasing diagnosis rates in asymptomatics by roll-out of rapid test can facilitate timely vaccination and testing, and potentially enhance the impact of gonorrhea vaccination programmes. Lastly, our study did not consider sentiments on vaccination, which was previously found to downwardly influence the public health impact of vaccination programmes [15].

## Conclusions

Vaccination of MSM against gonorrhea, according to risk or on diagnoses in sexual health clinics in Singapore, with the 4CMenB vaccine can be considered. Development of gonorrhea-specific vaccines should prioritize maximizing vaccine efficacy over duration of protection.

## Supporting information

**S1 Text. Supplementary information.**
(DOCX)

## Author contributions

**Conceptualization:** Lin Geng, Lilith K. Whittles, Jue Tao Lim.

**Data curation:** Lin Geng.

**Formal analysis:** Lin Geng.

**Funding acquisition:** Azra Ghani, Jue Tao Lim.

**Investigation:** Yihao Chen, Rayner Kay Jin Tan, Azra Ghani, Jue Tao Lim.

**Methodology:** Lin Geng, Lilith K. Whittles, Borame L. Dickens, Martin T. W. Chio, Rayner Kay Jin Tan, Jue Tao Lim.

**Project administration:** Lin Geng, Lilith K. Whittles, Martin T. W. Chio, Azra Ghani, Jue Tao Lim.

**Resources:** Lin Geng, Martin T. W. Chio, Jue Tao Lim.

**Software:** Lin Geng, Lilith K. Whittles.

**Supervision:** Lilith K. Whittles, Martin T. W. Chio, Rayner Kay Jin Tan, Azra Ghani, Jue Tao Lim.

**Validation:** Lilith K. Whittles, Borame L. Dickens, Martin T. W. Chio, Rayner Kay Jin Tan, Jue Tao Lim.

**Visualization:** Lin Geng.

**Writing – original draft:** Lin Geng, Jue Tao Lim.

**Writing – review & editing:** Lin Geng, Lilith K. Whittles, Borame L. Dickens, Martin T. W. Chio, Yihao Chen, Rayner Kay Jin Tan, Azra Ghani, Jue Tao Lim.

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
