## [Editor Report · Decision Letter 0]

5 Aug 2024

Dear Dr Geng,

Thank you for submitting your manuscript entitled "Potential public health impacts of gonorrhoea vaccination programmes under declining incidences: a modelling analysis" for consideration by PLOS Medicine.

Your manuscript has now been evaluated by the PLOS Medicine editorial staff and I am writing to let you know that we would like to send your submission out for external peer review.

Please re-submit your manuscript within two working days, i.e. by Aug 07 2024 11:59PM. Please do let us know if you need more time (ssunny@plos.org).

Kind regards,

Syba

Syba Sunny, MBBS, MRes, FRCPath

Associate Editor

PLOS Medicine

---

## [Decision Letter · Decision Letter 1]

18 Oct 2024

Dear Dr Geng,

Many thanks for submitting your manuscript "Potential public health impacts of gonorrhoea vaccination programmes under declining incidences: a modelling analysis" (PMEDICINE-D-24-02533R1) to PLOS Medicine. The paper has been reviewed by subject experts and a statistician; their comments are included below and can also be accessed here: [LINK]

As you will see, the value of your work was evident to many, though the reviewers did raise a number of points to address. After discussing the paper with the editorial team and an academic editor with relevant expertise, I'm pleased to invite you to revise the paper in response to the reviewers' comments. We plan to send the revised paper to some or all of the original reviewers, and we cannot provide any guarantees at this stage regarding publication.

We ask that you submit your revision by Nov 08 2024 11:59PM. However, if this deadline is not feasible, please contact me by email, and we can discuss a suitable alternative.

Don't hesitate to contact me directly with any questions (ssunny@plos.org).

Best regards,

Syba

Syba Sunny, MBBS, MRes, FRCPath

Associate Editor

PLOS Medicine

ssunny@plos.org

Comments from the academic editor:

The academic editor was supportive of your manuscript and wrote this: ‘The work is important and novel - when there is uncertainty in the field, mathematical models can provide insights into potential scenarios as outlined by the reviewers including the (fundamental) uncertainty of the efficacy of the meningococcal vaccine and GC resistance to antibiotics. Analyses can also inform the ideal product profile for gonorrhea vaccine development.’ She agreed with the reviewer comments and asked that you were given the opportunity to respond to them.

Comments from the reviewers:

Reviewer #1: In this paper, the author developed and calibrated a simulation model to examine the impact of various vaccination strategies on gonorrhea incidence. The model was specifically applied to the MSM population in Singapore. The author concluded that vaccinating MSM against gonorrhea based on risk assessment in sexual health clinics in Singapore is a viable approach, and that vaccine development efforts should prioritize maximizing efficacy over the duration of protection. The topic is innovative, and the findings make a significant contribution to the existing literature. I recommend that this paper be considered for publication after the following concerns are addressed.

1. In the model structure, the author assumed that all individuals entering treatment can be successfully treated. However, for gonorrhea transmission, particularly within MSM populations, antibiotic resistance is a critical issue. While the author noted that few resistant strains have been diagnosed in Singapore, I recommend that antibiotic resistance should be considered due to the uncertainty surrounding future transmission dynamics. I suggest adding a resistance compartment to the simulation model or conducting a sensitivity analysis on resistance scenarios, with a discussion of the corresponding results. It could be insightful to examine the impact of antibiotics not only on transmission but also on resistance. Incorporating this aspect would enhance the model's applicability to gonorrhea transmission in other regions and strengthen the overall conclusions.

2. In the discussion, the author mentioned that the cost-effectiveness of the vaccination strategies was not analyzed due to the lack of publicly available cost data on gonorrhea treatment. However, Li et al. (2022) estimated the attributable disease burden in terms of discounted lifetime costs and quality-adjusted life-years (QALYs) lost for the United States. The author may want to revise this section of the discussion to clarify this point and avoid potential confusion.

Reviewer #2: Thank you for the opportunity to review this paper. My comments are below.

Abstract (Minor comments)

In the methods section, revise "…reducing gonorrhea transmission in the men who have sex with men population…". Suggest changing to "…reducing gonorrhea transmission in men who have sex with men…"

In the methods section, delete "to" in the following sentences: "…offering vaccination to individuals attending to sexual health clinics for testing…"

Change "diagnoses" to "diagnosed" in the following sentence "…offering vaccination to individuals attending to sexual health clinics and were diagnoses with gonorrhea…".

In vaccination scenarios 2 and 3, the authors discuss offering vaccines to "individuals". Do the authors mean all persons attending a sexual health clinic (SHC) or MSM attending a SHC.

Does the diagnoses in vaccination scenario 4 represent any diagnoses at any time (ever diagnosed) or diagnoses in a prior time period eg diagnoses in past 12 months.

Please be clear - In the background section, the authors state that they explored the efficacy of potential GC vaccination in the context of declining GC incidence. In the methods, the authors state that "they employed an integrated transmission dynamic model...in reducing GC transmission in the men who have sex with men population in Singapore" but the results section describes the effect on MSM. Are the authors modeling the impact of vaccination on GC in the general population or MSM OR they vaccinating the general population in the six scenarios highlighted to assess the impact on MSM??? This is not very clear and confusing. The authors should clarify this so that the reader does not have to infer the objective of the paper.

First sentence of the Interpretation. "Vaccination of MSM against gonorrhea…" This is the first reference to MSM without spelling out what it means. The term "men who have sex with men" was first used in the methods but the reference to the abbreviation, MSM, was not made.

Introduction.

1st paragraph line 5. Change "asymptotic" to "asymptomatic".

1st paragraph, line 8. Change "urethral" to "urethra".

Paragraph 2: Authors should please list the "non-pharmaceutical interventions (NPIs)" that they are referring to. Are they referring to stay-at-home orders, lockdown that may have impacted sexual relationships or STI testing?

Paragraph 3 line 6. Suggest changing "…as of date.,." to "…to date…"

Paragraph 3 line 13. The authors state that 4CMenB has a 22% efficacy against GC and cite the Molina et al study (Doxycycline prophylaxis and meningococcal group B vaccine to prevent bacterial sexually transmitted infections in France (ANRS 174 DOXYVAC): a multicentre, open-label, randomised trial with a 2 × 2 factorial design. Lancet ID). However, this study did not detect a significant protective effect (aHR 0·78 [95% CI 0·60-1·01]; p=0·061) against GC for reasons Molina and colleagues clearly outline in their paper, such as a potentially underpowered sample size to detect a small effect size. It is not correct to state that this study suggested "…a more conservative 22% efficacy of MenB-4C vaccine in gonorrhea prevention…" when it was not significantly protective.

Last paragraph of introduction: Include a statement that clearly states the objective of this study. The last paragraph includes sentences that contain elements of the objective of their modelling study but it is left to the reader to try and decipher the study's objective.

Methods.

First paragraph, line 1: MSM is already abbreviated in the intro (para 1, line 6). There is no need to abbreviate again in the methods. Also, please delete "the" before men who have sex with (MSM) population in Singapore (paragraph 1, line 2).

Data

Are the 2023 GC incidence estimates of MSM from the communicable surveillance system of Singapore similar to the proxy estimates from 2004-2018 that the authors calculated. I realize male same sex behavior was decriminalized in 2022 in Singapore but it is important to know to determine if the incidence estimate used for this analysis has any bearing on current estimates from 2023.

Vaccination scenario:

Paragraph 2: In the VBE scenario, authors state they considered vaccination of male adolescents before they reach sexually active age but do not state what this age. Please include in the paper.

VoA Scenario 2: In VoA, authors offer vaccination to MSM when they attend a SHC. How do the authors define MSM in this scenario - men who report ever having sex with a man OR men who had sex with a man within a specified period (such as in the previous 12 months or some other time interval)? Please include this information in the paper.

Please check the confidence interval for the VbE strategy for the estimate of # of vaccinations administered (Table 4). The # of vaccinations administered, and the upper and lower limit of the confidence intervals are all 172701.

In the foot notes of Table 1, the authors state that the calculation for averted cases per dose is computed as the difference between # of treated GC patients between the baselines case and the respective vaccination strategy, divided by the # of vaccination doses. In Table 4, does this mean that the averted cases per dose (column 2) can be calculated as averted cases (column 2) divided by number of vaccinations administered (column 5). If this is interpretation is correct, the current calculation does not result in column 2 using the data in Table 4.

Vaccine profile. The authors used a vaccine effectiveness of 31% as the primary vaccine profile of 4CMenB vaccine this analysis (Table 2) and cite a study by Petousis-Harris et al. Effectiveness of a group B outer membrane vesicle meningococcal vaccine against gonorrhoea in New Zealand: a retrospective case-control study. Lancet). However, the Petousis-Harris study assessed the effectiveness of MenZB vaccine (a vaccine that is no longer available) against GC and not 4CMenB. Although the outer membrane protein components of 4CMenB are similar to MenZB, 4CMenB contains recombinant proteins that some research shows may provide additional protection over MenZB (Semchenko et al. 2019. The Serogroup B Meningococcal Vaccine Bexsero Elicits Antibodies to Neisseria gonorrhoeae). When Whittles and colleagues conducted their study, I concede that data of 4CMenB were limited but since then there are at least 7 studies on VE estimates of 4CMenB and 2 meta-analytical studies that have published estimated pooled VE estimates.

Results.

Para 1.line 4. Suggest changing "infectees" to "infected persons".

How did the authors estimate high risk population among MSM. What parameters were used? Whittles et al. 2022 used data from Genitourinary Medicine Clinic Activity Dataset and the Gonococcal Resistance to Antimicrobials Surveillance Programme. The authors of this paper used data from the communicable disease surveillance system of Singapore. How did the authors estimate the number of MSM to categorize as high-risk MSM (i.e. those that reported 5 or more partners annually) for this study (Table 2). Was this inferred from male or female incidence in the surveillance system? Please explain this similar to the way the authors estimated GC incidence estimate among MSM was explained.

Although describing the model is important, I would argue that the accuracy of input parameters are even more important as they inform the accuracy and interpretation of the findings, especially as data on MSM in Singapore is lacking.

I realize that the authors used a cut-off of 5 partners annually based on prior work by Whittles, 2022. However, did the authors consider other more proximal factors that may be suggestive of elevated sexual risk such as condomless anal sex with a casual partner, previous STD in past 12 months, HIV PrEP use (due (due to potential risk compensation behaviors), chemsex, or substance use history. Five sex partners in past 1 year by itself without context such as whether these are concurrent multiple partners or partners with whom they engaged in condomless anal sex does not necessarily speak to risk. MSM who report 5 main, non-concurrent, sex partners who are non-concurrent in the past year may in fact be at lower sexual risk than MSM who report 4 partners (1 main partner with three concurrent casual partners) in the past year. This is something to think about.

Discussion.

Para 1 line 3. Authors state "…strategies which targeted individuals on diagnoses…" I suspect that the authors mean MSM. I urge the authors to stick to MSM rather than using population or individuals when they in fact mean MSM. It is confusing and a bit hard to follow.

para 2:Do the authors think that the reduced efficiency of vaccination in VoA and VaR compared with VoD should play a role in informing vaccination recommendation strategies. In other words, does the efficiency of a vaccination strategy supersede the proportion of averted cases or vice versa?

General comment: The paper may require a minor bit of editing as there are some typos in the paper.

Reviewer #3: 1. There is limited information regarding the justification of some key parameters of compartment model, particularly those related to the mixing between high- and low-risk groups. While the model uses an assortativity parameter, it would be useful to provide sensitivity analyses for varying (ϵ/ epsilon) values. This could highlight how different mixing assumptions affect transmission dynamics.

2. The rate of transmission (β/beta) is modelled as a linear function over time (Equation 1 in the supplementary). However, real-world transmission dynamics are likely to be more complex. Have other functional forms (e.g., exponential decay) been tested? It would strengthen the paper to discuss why a linear trend was chosen and whether alternative forms were explored during model calibration.

3. The use of Markov chain Monte Carlo (MCMC) for model calibration is appropriate, but the description of the convergence diagnostics could be more detailed. Specifically, did the authors assess convergence using statistical tests (e.g., Gelman-Rubin diagnostics) or rely solely on visual inspection of trace plots? It would be helpful to provide quantitative convergence statistics in the supplementary material to support the claim of robust calibration.

4. While the model is calibrated to historical gonorrhoea incidence in Singapore, it is not clear whether any model validation was performed. The authors could strengthen their argument by splitting the dataset into calibration and validation sets, and then testing how well the calibrated model predicts the validation data.

5. The sensitivity analyses cover a good range of vaccine uptake (10%, 33%, 100%) and efficacy (22%, 31%, 47%) parameters. However, it would be useful to explore interactions between vaccine efficacy and duration of protection. For instance, does increasing efficacy reduce the importance of long-duration vaccines, or is there a synergistic effect between high efficacy and long duration?

6. Sensitivity analyses focus primarily on vaccination parameters. Since behavioural parameters (e.g., screening rates, risk behaviour) play a crucial role in transmission dynamics, including them in the sensitivity analyses would provide a more comprehensive understanding of model behaviour under different scenarios.

7. The model assumes that treated individuals (TjT) return to the uninfected state (Uj) with the same risk of infection as those who were never infected. This assumption overlooks the potential differences in behaviour, immunity, or vulnerability between previously infected and never-infected individuals. For example, individuals who are treated for gonorrhoea might have a different risk of reinfection due to changes in sexual networks, or the possibility of partial immunity from treatment.

8. While the model acknowledges the presence of asymptomatic cases in the population, it does not fully explore their impact on the overall transmission dynamics. Asymptomatic individuals may unknowingly spread the infection over extended periods, potentially making them significant drivers of the epidemic. If asymptomatic individuals are not screened or treated as often, their role in sustaining transmission could be underestimated.

9. The authors mention the use of Bayesian methods for parameter estimation but provide limited details on the prior distributions used for key parameters (e.g., transmission rate β\beta, recovery rate, vaccine efficacy). Additionally, sensitivity to different prior assumptions could be tested, especially if certain priors were based on weak or uncertain evidence.

10. The formulation of the force of infection (λj\lambda) is correct, but the description in the text could be clearer. Specifically, more detail could be provided on the implications of the assortative parameter (ϵ\epsilon) and the proportionate mixing assumption. Providing a plot or example of how varying assortative affects transmission dynamics would improve the reader's understanding.

11. The choice of a Negative Binomial distribution to model gonorrhoea case counts is sound, given the over-dispersion typically observed in such data. However, the shape parameter (κ\kappa) is introduced but not fully explained. The paper would benefit from further discussion on how κ\kappaκ was estimated or calibrated and whether alternative distributions (e.g., Poisson or zero-inflated models) were considered and compared.

12. The authors appropriately report 95% credible intervals for key results, but they could further elaborate on how uncertainty in different parameters (e.g., vaccine efficacy, behavioural parameters) contributes to overall uncertainty in the results. A decomposition of uncertainty by source would provide valuable insights into which assumptions or parameters have the most significant impact on model outcomes.

13. The model distinguishes between high and low sexual activity groups but assumes homogeneity within each group. In reality, there might be significant heterogeneity in both behaviour and transmission risks within these groups. A discussion on how heterogeneity within groups could impact model results would be valuable, and considering a more granular approach (e.g., splitting the high-risk group into further subgroups) could yield different insights.

14. The model assumes a constant population size over time, with individuals entering and exiting the sexually active population. While this is a common assumption in transmission models, it would be useful to acknowledge potential demographic changes (e.g., shifts in the age distribution, migration) and how they might influence the results. Sensitivity analysis based on varying population growth rates could provide further insights into the model's robustness.

15. The model assumes a constant natural recovery rate for asymptomatic infections. It might be worth exploring if recovery rates vary based on factors like the time since infection, health status, or co-infections. Additionally, it would be helpful to mention if there is any empirical evidence that supports the assumed constant recovery rate, or if any variability in this parameter was tested during sensitivity analysis.

16. The paper presents a screening rate equation (Equation 2 in the supplementary), which assumes a fixed relationship between the high- and low-risk groups (ω\omega). However, screening rates could vary over time based on factors like public health campaigns, changes in healthcare access, or risk perception. Modeling time-varying screening rates, or at least discussing their potential influence, would enhance the realism of the model.

17. The manuscript mentions the use of proxies for MSM incidence due to unavailable data (Supplementary Information, Section 2.1). While this approach is reasonable, it would be beneficial to provide more detail on how uncertainty due to these data limitations is incorporated into the model. Specifically, if the authors explored the impact of different proxy assumptions on the model outputs, it would demonstrate the robustness of the conclusions to data-related uncertainty.

18. The manuscript provides a good analysis of how vaccine efficacy and duration of protection influence outcomes. However, there is limited discussion on how realistic these assumptions are. Are the vaccine characteristics assumed in the model (e.g., 31% efficacy, 4-year duration of protection) based on real-world clinical data, or are they optimistic estimates? Including a section on how these values compare to available empirical data on vaccine performance (both for meningococcal and potential gonorrhoea vaccines) would ground the results in reality.

19. The authors could enhance their interpretation by comparing their results to similar models used for other sexually transmitted infections (e.g., HPV, chlamydia). This comparison would provide context to the findings, particularly regarding the efficiency of different vaccination strategies in low-prevalence settings. Such a comparison might also highlight generalizable lessons or key differences in vaccination program design across diseases.

20. The model is projected over a 10-year period. This time frame may not fully capture the longer-term effects of vaccination programs, especially for interventions like adolescent vaccination (VbE) that may have delayed benefits. Extending the model to 20 or 30 years could provide insights into whether long-term herd immunity can be achieved or if the benefits of certain strategies increase over time.

21. While the authors present credible intervals for key results, some figures could benefit from a clearer representation of uncertainty. For example, Figures 2 and 3 use box plots and radar graphs to display outcomes, but adding additional uncertainty visualization (such as density plots or violin plots) might make the distributions of outcomes more apparent.

22. It appears that the table currently labelled Table S3 in the supplementary file, which presents "Fitted transmission parameters: notation, definition, prior distribution, parameter bounds, and posterior estimates," should actually be Table S4 based on the content flow. It seems that there are other tables (e.g., Tables S1, S2) preceding it. Therefore, the numbering of Table S3 should be revised to Table S4, and the subsequent tables should be renumbered accordingly to maintain consistency.

23. The graphical representation of vaccination strategy impact (Figure 2) is effective. However, some of the efficiency metrics (e.g., averted cases per dose) could benefit from a clearer legend and explanation in the figure captions. Additionally, radar charts are visually appealing but may obscure some important nuances in the data.

---

* Please upload any figures associated with your paper as individual TIF or EPS files with 300dpi resolution at resubmission; please read our figure guidelines for more information on our requirements: http://journals.plos.org/plosmedicine/s/figures. While revising your submission, please upload your figure files to the PACE digital diagnostic tool, https://pacev2.apexcovantage.com/. PACE helps ensure that figures meet PLOS requirements. To use PACE, you must first register as a user. Then, login and navigate to the UPLOAD tab, where you will find detailed instructions on how to use the tool. If you encounter any issues or have any questions when using PACE, please email us at PLOSMedicine@plos.org.

* At this stage, we ask that you include a short, non-technical Author Summary of your research to make findings accessible to a wide audience that includes both scientists and non-scientists. The Author Summary should immediately follow the Abstract in your revised manuscript. This text is subject to editorial change and should be distinct from the scientific abstract. Ideally each sub-heading should contain 2-3 single sentence, concise bullet points containing the most salient points from your study. In the final bullet point of 'What Do These Findings Mean?', please include the main limitations of the study in non-technical language. Please see our author guidelines for more information: https://journals.plos.org/plosmedicine/s/revising-your-manuscript#loc-author-summary (Please remove the ‘Research in context’ section.)

FIGURES AND TABLES

SUPPLEMENTARY MATERIAL

REFERENCES

MODELLING STUDIES

The following list is derived from Geoffrey P Garnett, Simon Cousens, Timothy B Hallett, Richard Steketee, Neff Walker. Mathematical models in the evaluation of health programmes. (2011) Lancet DOI:10.1016/S0140-6736(10)61505-X: 

* If pertinent, please provide a diagram that shows the model structure, including how the natural history of the disease is represented, the process and determinants of disease acquisition, and how the putative intervention could affect the system.

* Please provide a complete list of model parameters, including clear and precise descriptions of the meaning of each parameter, together with the values or ranges for each, with justification or the primary source cited and important caveats about the use of these values noted.

* Please provide a clear statement about how the model was fitted to the data, including goodness-of-fit measure, the numerical algorithm used, which parameter varied, constraints imposed on parameter values, and starting conditions.

* For uncertainty analyses, please state the sources of uncertainties quantified and not quantified [can include parameter, data, and model structure].

* Please provide sensitivity analyses to identify which parameter values are most important in the model. Uncertainty estimates seek to derive a range of credible results on the basis of an exploration of the range of reasonable parameter values. The choice of method should be presented and justified.

* Please discuss the scientific rationale for the choice of model structure and identify points where this choice could influence conclusions drawn. Please also describe the strength of the scientific basis underlying the key model assumptions.

* For studies that develop a prediction model or evaluate its performance, please ensure that the study is reported according to the TRIPOD statement (https://www.equator-network.org/reporting-guidelines/tripod-statement) and include the completed checklist as Supporting Information. Please add the following statement, or similar, to the Methods: "This study is reported as per the Transparent Reporting of a Multivariable Prediction Model for Individual Prognosis Or Diagnosis (TRIPOD) statement (S1 Checklist)." For studies using machine learning, please use the TRIPOD-AI checklist. When completing the checklist, please use section and paragraph numbers, rather than page numbers.

---

## [Decision Letter · Decision Letter 2]

23 Dec 2024

Dear Dr. Geng,

Thank you very much for re-submitting your manuscript "Potential public health impacts of gonorrhoea vaccination programmes under declining incidences: a modelling analysis" (PMEDICINE-D-24-02533R2) for review by PLOS Medicine. I am writing on behalf of my colleague Dr. Sunny.

I have discussed the paper with my colleagues and the academic editor and it was also seen again by 2 of the original reviewers. I am pleased to say that provided the remaining editorial and production issues are dealt with we are planning to accept the paper for publication in the journal.

[LINK]

We look forward to receiving the revised manuscript by Dec 30 2024 11:59PM.   

Sincerely,

Alison Farrell, PhD

Senior Editor 

PLOS Medicine

plosmedicine.org

Requests from Editors:

We ask that you address the remaining comments from Reviewer 3 in the main text and discussion, expand the limitations discussion to address their concerns, and clarify where limited data availability may have impacted GC incidence estimates.

Please also address the following text edits:

Abstract:

line 16: designed to protect against Neisseria meningitidis infection

line 23: should this be 'acquisition and transmission' or only 'transmission'?

lines 26,27: offering vaccination to individuals attending sexual health clinics and who were diagnosed with gonorrhoea

line 31: delete 'during'

line 37: what is the intervention time?

line 39: please correct phrasing ' but three times efficient higher'

line 45: replace which with that

line 51: can be considered for what? Please qualify and more clearly articulate the study implications.

lines 51, 52: The abstract does not focus on duration of protection, so why does the concluding sentence? Please revise sentence, and/or provide a sentence in the results section of the abstract pertaining to this issue. I suggest focusing the last sentence on the development of gonorrhoea-specific vaccines and the design and implementation of vaccination programmes to maximise their protective efficacy and public health impact.

line 57: Vaccination of

line 61: use of "can confer" seems too strong given that there are no approved gonorhhea-specific vaccines. Please consider revising phrasing. E.g. n. Gonorrhoea-specific vaccines may confer greater public health impacts by maximising efficacy over the duration of protection. Separately, vaccine developers typically prioritize efficacy, therefore is this qualification (relative to duration) necessary, versus focusing on the implementation of targeted vaccination programmes, which the study models?

The manuscript contains grammatical errors throughout. Please correct.

Introduction

lines 95-97: please include in this sentence the pathogen targeted by MenB.

line 102: here and throughout, please use lower case g for gonorrhoea

lines 118, 119: 'potential' used 2x. Please remove one use.

line 123: add 'among MSM'? Also mention the time range(s) of the analyses and modelling?

Discussion

replace monkeypox with mpox

Please start paragraph discussing study limitations with a sentence to that effect.

Please directly state in the text that there aren’t publicly available gonorrhea incidence estimates for Singapore beyond 2020 (as per your reply to reviewer 3)

We also ask you to make the following formatting edits to the manuscript:

* It appears that one or more study authors is affiliated with one or more of the agencies that funded the study. Thus, the statement “The funders had no role in study design, data collection and analysis, decision to publish, or preparation of the manuscript” does not apply. Please revise the Financial Disclosure accordingly, as in "[Author name] is [author's role] at [funding agency]. The funders had no other role in study design…..”

Please expand upon your data availability statement.

"* The Data Availability Statement (DAS) requires revision. For each data source used in your study:

* Please review your manuscript and edit to ensure compliance with our inclusive language requirements https://journals.plos.org/plosmedicine/s/human-subjects-research#loc-categorization

* Please define all elements of box plots in the figure caption - center line, box limits and whiskers.

* Please confirm that your title complies with to PLOS Medicine's style. Your title must be nondeclarative and not a question. It should begin with main concept if possible. "Effect of" should be used only if causality can be inferred, i.e., for an RCT. Please place the study design ("A randomized controlled trial," "A retrospective study," "A modelling study," etc.) in the subtitle (ie, after a colon).

Comments from Reviewers:

Reviewer #1: Thank you to the editors and reviewers, as well as the author for addressing all the comments. I agree that this manuscript has been significantly improved and is now suitable for publication.

— Xuecheng Yin

Reviewer #3: I appreciate the authors' comprehensive revisions and the effort to address the concerns raised during the initial review. The manuscript is much improved, with additional sensitivity analyses, clarified justifications, and expanded discussions on key modelling assumptions. The responses were well-detailed and thoughtful.

Below are some additional comments that, if addressed, could further strengthen the manuscript.

1. While the authors provided a rationale for choosing a linear functional form, it would add value to briefly discuss the implications of alternative forms in the results or limitations section. For example, if exponential decay was tested but found less fitting, highlighting why it was inappropriate would strengthen the justification.

2. The model does not currently account for feedback loops, such as changes in sexual risk behaviour due to perceived protection from vaccination. A discussion on how these unmodeled dynamics might impact real-world outcomes would be valuable.

3. Including a discussion on how different public health strategies (e.g., intensified outreach, rapid diagnostics) could interact with vaccination to enhance impact might make the findings more actionable.

4. Although the revised legends are improved, further clarity on metrics (e.g., how averted cases per dose are calculated) in the main text would enhance the utility of figures for the reader.

5. While sensitivity analyses were conducted, a summary or visualization of how varying assortativity impacts key outcomes (e.g., averted cases, vaccination efficiency) could make the implications more accessible.

6. Ensure the most recent studies on vaccine efficacy (e.g., Abara et al., 2024) are integrated seamlessly into the discussion, particularly in comparison with older studies.

[LINK]

---

## [Editor Report · Decision Letter 3]

7 Jan 2025

Dear Dr Geng, 

On behalf of my colleagues and the Academic Editor, Philippa Dodd, I am pleased to inform you that we have agreed to publish your manuscript "Potential public health impacts of gonorrhoea vaccination programmes under declining incidences: a modelling study" (PMEDICINE-D-24-02533R3) in PLOS Medicine.

Please also note the following:

Infection in the first sentence of the abstract needs to be singular.

In authors' response to reviewer 3 in the amended discussion statement:"We hypothesize that vaccination may be accompanied by a shift towards healthier sexual behavioral practices among the study population, and thereby increase the public health impact of vaccination programmes", the authors actually should consider that vaccination may be accompanied by higher risk activities due to an assumption of efficacious vaccine-mediated protection. I believe this is what the reviewer was suggesting that you consider.

PRESS

Sincerely, 

Alison Farrell, Ph.D. 

Senior Editor 

PLOS Medicine